# Non-Oncogene Addiction of *KRAS*-Mutant Cancers to IL-1β via Versican and Mononuclear IKKβ

**DOI:** 10.3390/cancers15061866

**Published:** 2023-03-20

**Authors:** Magda Spella, Giannoula Ntaliarda, Georgios Skiadas, Anne-Sophie Lamort, Malamati Vreka, Antonia Marazioti, Ioannis Lilis, Eleni Bouloukou, Georgia A. Giotopoulou, Mario A. A. Pepe, Stefanie A. I. Weiss, Agnese Petrera, Stefanie M. Hauck, Ina Koch, Michael Lindner, Rudolph A. Hatz, Juergen Behr, Kristina A. M. Arendt, Ioanna Giopanou, David Brunn, Rajkumar Savai, Dieter E. Jenne, Maarten de Château, Fiona E. Yull, Timothy S. Blackwell, Georgios T. Stathopoulos

**Affiliations:** 1Department of Physiology, Faculty of Medicine, University of Patras, 26504 Rio, Greece; 2Comprehensive Pneumology Center and Institute for Lung Biology and Disease, Helmholtz Center Munich-German Research Center for Environmental Health, 81377 Munich, Germany; 3Research Unit Protein Science-Core Facility Proteomics, Helmholtz Center Munich–German Research Center for Environmental Health, 80939 Munich, Germany; 4Center for Thoracic Surgery Munich, Ludwig-Maximilians-University of Munich and Asklepios Medical Center, 82131 Gauting, Germany; 5Department of Internal Medicine V, Ludwig-Maximilian-University of Munich, 81377 Munich, Germany; 6Max Planck Institute for Heart and Lung Research, 61231 Bad Nauheim, Germany; 7Frankfurt Cancer Institute (FCI), Goethe University, 60596 Frankfurt am Main, Germany; 8Department of Internal Medicine and Institute for Lung Health (ILH), Justus Liebig University, 35392 Giessen, Germany; 9Max-Planck-Institute of Neurobiology, 82152 Planegg, Germany; 10Buzzard Pharmaceutical, 17165 Stockholm, Sweden; 11Department of Pharmacology, Vanderbilt University School of Medicine, Nashville, TN 37240, USA; 12Department of Medicine, Division of Allergy, Pulmonary and Critical Care Medicine, Vanderbilt University School of Medicine, Nashville, TN 37240, USA

**Keywords:** nuclear factor-κB, interleukin-1β, cancer, inflammation, non-oncogene addiction, bioluminescence

## Abstract

**Simple Summary:**

*Kirsten rat sarcoma virus (KRAS)*-mutant cancers are frequent, metastatic, lethal, and largely undruggable. The aim of this study was to investigate the pathways through which *KRAS*-mutant cancers foster their growth, thereby unravelling novel therapeutic targets. We show that *KRAS*-mutant tumors secrete the protein versican, which then drives the activation of NF-κB kinase (IKK) β in a type of host immune cells called macrophages. Following this activation, macrophages fuel the tumor with interleukin (IL)-1β, to close an inflammatory loop through which *KRAS*-mutant cancers attract host immune cells to the tumor site to accelerate tumor growth and aggressiveness. Importantly, we show that targeting IL-1β and/or versican can be an effective treatment for *KRAS*-mutant cancers, holding great promise for cancer patients.

**Abstract:**

*Kirsten rat sarcoma virus (KRAS)*-mutant cancers are frequent, metastatic, lethal, and largely undruggable. While interleukin (IL)-1β and nuclear factor (NF)-κB inhibition hold promise against cancer, untargeted treatments are not effective. Here, we show that human *KRAS*-mutant cancers are addicted to IL-1β via inflammatory versican signaling to macrophage inhibitor of NF-κB kinase (IKK) β. Human pan-cancer and experimental NF-κB reporter, transcriptome, and proteome screens reveal that *KRAS*-mutant tumors trigger macrophage IKKβ activation and IL-1β release via secretory versican. Tumor-specific versican silencing and macrophage-restricted IKKβ deletion prevents myeloid NF-κB activation and metastasis. Versican and IKKβ are mutually addicted and/or overexpressed in human cancers and possess diagnostic and prognostic power. Non-oncogene *KRAS*/IL-1β addiction is abolished by IL-1β and TLR1/2 inhibition, indicating cardinal and actionable roles for versican and IKKβ in metastasis.

## 1. Introduction

Tumor-associated inflammation is intimately linked with tumor progression and therapy response [1]. Interleukin (IL)-1β is an important mediator of tumor-associated inflammation and its inhibition via the monoclonal antibody canakinumab was recently shown to possess strong protective effects against incident lung cancer in an exploratory analysis of the canakinumab anti-inflammatory thrombosis outcomes study (CANTOS) [2]. Unexpectedly, the phase III CANOPY-2 trial (ClinicalTrials.gov NCT03626545) investigating second/third-line canakinumab with docetaxel against non-small cell lung cancer (NSCLC), irrespective of histologic subtype and driver mutation, was negative for unknown reasons [3]. To this end, the protective effects of canakinumab in the CANTOS trial NSCLC exploratory study were significantly stronger for current and former smokers and for incipient lung adenocarcinoma (LUAD) histologic subtype, with both carrying high mutation rates of the Kirsten rat sarcoma virus (*KRAS*) proto-oncogene GTPase (encoded by the *KRAS/Kras* genes in humans/mice).

Multiple lines of evidence dictate that tumor genomic alterations largely define tumor-associated inflammation and the efficacy of immune-directed therapies [1]. To this end, NSCLC with high mutation burden and a smoking-associated trinucleotide signature were found to display more favorable and durable responses to the immune checkpoint inhibitor pembrolizumab targeting programmed cell death-1 (PD-1) [4]. Moreover, *STK11/LKB1* alterations were reported to be cardinal drivers of primary resistance to PD-1 inhibitors in *KRAS*-mutant LUAD, the most frequent and lethal histologic subtype of NSCLC [5]. Experimental evidence supports that the immune landscape and vulnerabilities of various tumor types can rely on single mutated driver oncogenes such as *KRAS* and *MYC* that orchestrate distinct transcriptional programs, dictate a tumor’s specific pro-inflammatory mediator secretory profile, and largely define the cellular composition of the tumor microenvironment [6,7]. In this regard, oncogenic *KRAS* is known to cooperate with pro-inflammatory nuclear factor (NF)-κB signaling in cancer cells to drive stemness, pro-inflammatory mediator elaboration, and responsiveness to IL-1β signaling [8,9,10,11], and is ideally positioned as a biomarker of therapeutic response to anti-IL-1β therapy.

Here, we show that *KRAS*-mutant cancers display specific non-oncogene addiction to host-provided IL-1β in humans and mice. We further elucidate how these tumors activate NF-κB in tumor-associated macrophages in order to elicit the IL-1β they require for sustained growth. Mutant *KRAS*-IL-1β addiction is mediated via secretion of the glycoprotein versican (VCAN) by tumor cells, which inhibits the NF-κΒ kinase (IKK) β in macrophages, resulting in IL-1β release into the tumor microenvironment. Importantly, the VCAN-IKKβ axis is shown to be required for the sustained growth of *KRAS*-mutant tumors and to constitute a diagnostic and prognostic biomarker of these tumors. Finally, we show how non-oncogene addiction of *KRAS*-mutant cancers to myeloid IL-1β can be abolished by pharmacologic inhibition of IL-1β or the VCAN target toll-like receptor (TLR) 2. Our findings can be directly translated to, and tested, in clinical trials of IL-1β inhibition against genomically stratified LUAD.

## 2. Materials and Methods

### 2.1. Murine and Human Study Approval

All mice used for these studies were bred at the Department of Medicine of the University of Patras, Greece. Experiments were prospectively approved by the Veterinary Administration of the Prefecture of Western Greece (approval #276134/14873/2) and were conducted according to the European Union Directive 2010/63/EU [12]. Male and female experimental mice were sex-, weight (20–25 g)-, and age (6–12 weeks)-matched. Exact sample sizes (*n*) are included in the figures and their legends. Animals were assigned to experimental groups by randomization (when *n* ≥ 20) or alternation (when *n* < 20) with controls and experimental mice always being littermates, and transgenic animals enrolled case-control-wise. Data were collected by at least two blinded investigators from samples coded by non-blinded investigators. The Munich lung adenocarcinoma and Patras pleural effusion [13,14] clinical studies were conducted in accordance with the Helsinki Declaration [15], were approved by the Ludwig-Maximilians-University Munich Ethics Committee (approval #623-15) and the University of Patras Ethics Committee (approval #22699/21.11.2013), were registered with the German Clinical Trials Register (Deutsches Register Klinischer Studien; #DRKS00012649 [16]) and with ClinicalTrials.gov (Using pleural effusions to diagnose cancer; NCT03319472 [17]), respectively, and written informed consent was prospectively obtained from all patients. 

### 2.2. Reagents 

D-Luciferin potassium salt CAS# 115144-35-9 was from Biosynth (Lake Constance, Switzerland). Clodronate (Dichloromethylenediphosphonic acid disodium salt, CAS# 22560-50-5) and Hoechst 33258 nuclear dye (CAS# 23491-45-4), were from Sigma-Aldrich (St. Louis, MO, USA). Egg-phosphatidylcholine (CAS# 97281-44-2) was from Avanti Polar Lipids (Alabaster, AL, USA). Lentiviral shRNA, puromycin (CAS# 58-58-2) and lipopolysaccharide (LPS; catalogue # sc-3535) were from Santa Cruz (Dallas, TX, USA). Geneticin (G418; catalogue # 10131035) was from Thermo Fisher Scientific (Waltham, MA, USA). Recombinant human active versican (VCAN; catalogue # RPB817Mu01) and osteopontin (secreted phosphoprotein 1, SPP1; catalogue # APA899Hu61) were from Cloud-Clone Corp (Houston, TX, USA) and all other recombinant proteins were from Immunotools (Friesoythe, Germany). Bortezomib (CAS# 179324-69-7) was from Selleckchem (Houston, TX, USA). IL-1β ELISA (catalogue # 900-K47) was from Peprotech (London, UK). Primers were from VBC Biotech (Vienna, Austria). Isunakinra (EBI-005), a recombinant protein that binds to the interleukin-1 receptor 1 (IL1R1) and potently blocks IL-1α and IL-1β beta [18], was from Buzzard Pharmaceutical (Stockholm, Sweden), and the TLR1/TLR2 antagonist Cu-CPT22 or 3,4,6-Trihydroxy-2-methoxy-5-oxo-5H-benzocycloheptene-8-carboxylic acid hexyl ester (CAS# 1416324-85-0) [19] was from Merck (Darmstadt, Germany). Primers and lentiviral shRNA pool sequences are listed in Appendix A and antibodies in the respective methods sections.

### 2.3. Cells 

Lewis lung carcinoma (LLC, RRID:CVCL_4358), B16F10 skin melanoma (male, RRID:CVCL_0159), and PAN02 pancreatic adenocarcinoma cells (male, RRID:CVCL_D627) were from the National Cancer Institute Tumor Repository (Frederick, MD, USA). RAW264.7 murine myelomonocytic leukaemia (male, RRID:CVCL_0493) and mouse lung epithelial 12 (MLE12, female, RRID:CVCL_3751) cells were from ATCC (Manassas, VA). MC38 colon adenocarcinoma (female, RRID: CVCL_B288) and AE17 mesothelioma (female, RRID:CVCL_4408) cells were gifts from Dr. Barbara Fingleton (Vanderbilt University, Nashville, TN, USA) and Dr. Y.C. Gary Lee (University of Western Australia, Perth, Australia), respectively. *FVB* urethane-induced lung adenocarcinoma (FULA1) cells were produced in our laboratories (female, RRID: CVCL_A9KV). The cells were cultured at 37 °C in 5% CO_2_-95% air using Dulbecco’s modified Eagle’s medium (DMEM) supplemented with 10% fetal bovine serum, 2 mM L-glutamine, 1 mM pyruvate, 100 U/mL penicillin, and 100 mg/mL streptomycin. For in vivo injections, the cells were trypsinized, incubated with Trypan blue, counted with a grid hemocytometer according to the Neubauer method, and only 95% viable cells were used for the experiments. All in vitro experiments were repeated independently at least three times and the stated *n* always reflects the biological and not technical sample size. All cell lines have been repeatedly reported, were re-sequenced for *Kras* mutations and their status was verified to be the same as previously reported, and were tested annually for identity by short tandem repeats and for *Mycoplasma* spp. by PCR using primers GGGAGCAAACAGGATTAGATACCCT and TGCACCATCTGTCACTCTGTTAACCTC (amplicon size 270 bp) [10,11,20,21,22,23].

### 2.4. Experimental Mice 

*NGL* and *HLL* NF-κΒ reporter mice are described elsewhere [24,25]. Mice obtained from Jackson Laboratories (Bar Harbor, Lake Shore, MN, USA) were wild–type (*WT*) *C57BL/6J* mice (*C57BL/6*; #000664), *B6.129(Cg)-Gt(ROSA)26Sor^tm4(ACTB-tdTomato,-EGFP)Luo^/J* dual membranous fluorescent *Cre*-recombinase reporter mice (*mT/mG*; #007676) [26], *B6.129P2-Lyz2^tm1(cre)Ifo^/J* mice that express *Cre*-recombinase under control of the *Lyz2* promoter (*Lyz2.Cre*; #004781) [21,27], *B6.129P2-Gt(ROSA)26Sor^tm1(DTA)Lky^/J* mice that express Diphtheria toxin upon *Cre*-mediated recombination that results in cell suicide (*Dta*; #009669) [21,28], and *B6;129S-Tnf^tm1Gkl^/J Tnf*-deficient mice (*Tnf−/−*; #005540) [11,29]. B6.B4B6-*Chuk^<tm1Mpa>^*/Cgn (*Chuk^f/f^*) and B6.B4B6-*Ikbkb^<tm2.1Mpa>^*/Cgn (*Ikbkb^f/f^*) mice that carry conditional *Chuk* and *Ikbkb* alleles that are deleted upon *Cre*-recombinase expression [30,31], as well as *Il1b^tm1Yiw^ Il1b*-deficient mice (*Il1b−/−*; MGI #215739631) [32] and *Cpa3.Cre+/–* mast cell-deficient mice in which mast cells undergo *Trp53*-mediated apoptosis (*Cpa3.Cre*) [33] were described elsewhere and were kindly donated by their founders. All mice used for these studies were originated from or back-crossed > F12 generations to the *C57BL/6* background. For these studies, *n* = 929 mice were used. 

### 2.5. Mouse Tumor Models

For the generation of solid tumors, mice were injected subcutaneously (s.c.) in the shaven rear flank dermis with 5 × 10^5^ tumor cells in 100 μL of phosphate-buffered saline (PBS), as described elsewhere [10,11,20,22]. Mice were weekly examined for tumor volume (V) by measuring three vertical tumor diameters (d1, d2, d3) using the formula *V = π* × *d1* × *d2* × *d3* and were killed when the tumor volume reached 1 cm^3^ (PANO2 cells) or 2 cm^3^ (all other cell lines). For the induction of malignant pleural effusions (MPE), mice received intrapleural injections of 2 × 10^5^ cancer cells suspended in 100 μL PBS and were sacrificed when showing signs of sickness or at the time-points indicated (14–28 days post-tumor cell delivery depending on the cell line used) [10,11,22]. In all models, both the mice and the inoculated cancer cells were always syngeneic to avoid inflammatory allograft rejection and artificial NF-κB activation. 

### 2.6. Bioluminescence and Biofluorescence Imaging 

Mice were imaged for NF-κB reporter bioluminescent signal daily starting at day 10 post-tumor cell injection until sacrifice. For this, mice were anesthetized by isoflurane inhalation and were imaged for bioluminescence on a Xenogen Lumina II (PerkinElmer, Waltham, MA, USA) 5–20 min after delivery of 1 mg D-Luciferin potassium salt diluted in 100 μL of sterile water into a retro-orbital vein. Pleural tumors isolated from *NGL* mice were also imaged ex vivo for green biofluorescence using 410–440 nm background control excitation, 445–490 nm experimental excitation, and 515–575 nm emission passbands on a Xenogen Lumina II. Cells were imaged for bioluminescence on a Xenogen Lumina II 0, 4, 8, 16, and 24 h after a single addition of 300 μg/mL (equivalent to 1 mM) D-luciferin to the culture media. Data were analyzed using Living Image v.4.2 (PerkinElmer, Waltham, MA, USA) as described previously [10,11,20,21,22,34].

### 2.7. Sequencing 

Genomic DNA was extracted from cell lines using the GenElute Mammalian Genomic DNA Miniprep Kit (Sigma-Aldrich). *Kras* exons 1–3 were amplified by PCR using Phusion Polymerase (New England Biolabs, Ipswich, MA, USA) and 60 °C annealing temperature. Primers are described in Appendix A. PCR products were analyzed on 1% agarose gels, purified by QIAquick Gel Extraction Kit (Qiagen, Hilden, Germany) and sequenced by Eurofins Genomics (Ebersberg, Germany).

### 2.8. Constructs and Transfections

Control shRNA (sh*C*, sc-108080-V; target sequences are proprietary of the manufacturer) and anti-mouse *Vcan* shRNA (sh*Vcan*, sc-41904–V) pools were from Santa Cruz. The p*NGL* construct and lentiviral shRNA pools for the silencing of *Kras*, *Chuk*, *Ikbkb*, *Ikbke*, and *Tbk1* were described previously [11]. Lentiviral shRNA catalog numbers and target sequences are listed in Appendix A. For stable plasmid transfections, 10^5^ RAW264.7 cells were transfected with 5 μg DNA using Xfect (Takara, Mountain View, CA, USA), followed by selection by G418 (400–800 μg/mL). For stable shRNA transfection, 10^5^ tumor cells were transfected with lentiviral particles, and clones were selected by puromycin (2–10 μg/mL) [10,11].

### 2.9. Intrapleural Catheter 

For in vivo MPE drainage, a 1.2 cm-long catheter bearing serial fenestrations at 1 mm intervals was used, according to the detailed model description reported previously [35]. Mice were anesthetized using isoflurane and the catheter insertion site was shaved and disinfected using 70% ethanol and 10% povidone iodide, and the catheter was then installed into the pleural space and sutured under the skin. Mice were imaged pre- and post-MPE drainage and were sacrificed thereafter. 

### 2.10. Cytology 

MPE fluid was treated with red blood cell lysis buffer (155 mM NH_4_Cl, 12 mM NaHCO_3_, 0.1 mM EDTA) and MPE cells were centrifuged and stained with May-Grünwald-Giemsa. Slides were then mounted with Entellan (Merck Millipore, Darmstadt, Germany) and microscopically analyzed for the differential counting of pleural cells. Pleural lavage was performed by injecting 1 mL of saline intrapleurally and recovering it after 30 s. Pleural cells were enumerated with a haemocytometer, were centrifuged, were stained with May-Grünwald-Giemsa or with anti-rabbit F4/80 antibody (ab111101; Abcam, London, UK; RRID:AB_10859466) and hematoxylin, and were microscopically analyzed for the differential counting of pleural cells.

### 2.11. Flow Cytometry 

Pleural effusion cells were treated with red blood cell lysis buffer (155 mM NH_4_Cl, 12 mM NaHCO_3_, 0.1 mM EDTA), enumerated, and 0.5–1.0 × 10^6^ cells were processed for antibody staining. Pleural tumors were dissociated using 70 μm strainers (BD Bioscience, San Jose, CA, USA), enumerated, and 0.5–1.0 × 10^6^ cells were processed for antibody staining. BMDM were enumerated and 0.5–1.0 × 10^6^ cells were processed for antibody staining. All of the samples were suspended in 50 μL PBS with 2% FBS and 0.1% NaN_3_, and stained with the following antibodies: anti-CD45 (11-0451-85; eBioscience, Santa Clara, CA, USA; RRID:AB_465051), anti-CD11b (12-0112-82; eBioscience; RRID:AB_2734869), anti-Ly6C (45-5932-82; eBioscience; RRID:AB_2723343), anti-F4/80 (123128; Biolegend, San Diego, CA, USA; RRID:AB_893484), anti-Ly6G (127624; Biolegend; AB_10640819), anti-GFP eFluor^®^ 660 (50-6498-82; eBioscience; RRID:AB_11043268), anti-MHC Class II (17-5321; eBioscience; RRID:AB_469454), Alexa Fluor^®^ 647 anti-CD206 (141712; Biolegend; RRID:AB_10900420), biotinylated anti-firefly Luciferase (ab634; Abcam, London, UK; RRID:AB_305434), and streptavidin (17-4317-82; eBioscience), for 20 min in the dark at a concentration of 0.1 μg/10^6^ cells. Samples were analyzed on a CyFlowML flow cytometer using the FloMax Software (Partec, Darmstadt, Germany; RRID:SCR_014437), Flowing Software v.2.5.1 ([36]; RRID:SCR_015781) and FlowJo Software v10.6.2 (BD Bioscience, San Jose, CA, USA; RRID:SCR_008520). 

### 2.12. Immunohistochemistry 

For dark field immunofluorescence, pleural tumors were fixed in 4% paraformaldehyde overnight at 4 °C, cryoprotected with 30% sucrose, embedded in Tissue-Tek (Sakura, Tokyo, Japan) and stored at −80 °C. Cryosections of 10 μm were then post-fixed in 4% paraformaldehyde for 10 min, treated with 0.3% Triton X-100 for 5 min, blocked for 1 h in 1× phosphate-buffered saline (PBS) containing 10% fetal bovine serum (FBS), 3% bovine serum albumin (BSA), and 0.1% Tween 20, and then incubated with the indicated primary antibodies overnight at 4 °C. Sections were subsequently treated with fluorescent secondary antibodies, counterstained with Hoechst 33258 (CAS# 23491-45-4) and mounted with Mowiol 4-88 (Calbiochem, Darmstadt, Germany; CAS# 9002-89-5). The following primary antibodies were used: mouse anti-GFP (1:200 dilution; sc-9996; Santa Cruz, Dallas, TX, USA; RRID:AB_627695), rat anti-CD68:Alexa Fluor^®^ 488 (MCA1957A488T; AbD Serotec, Kidlington, UK; RRID:AB_1102282), mouse anti-CD45 FITC (11-0451-85; eBioscience; RRID:AB_465051), and rabbit anti-PCNA (1:3000 dilution; ab18197; Abcam, London, UK; RRID:AB_444313). Alexa Fluor donkey anti-mouse 488 (A21202; RRID:AB_141607), Alexa Fluor goat anti-rat 568 (A11077; RRID:AB_141874), and Alexa Fluor donkey anti-rabbit 568 (A10042; RRID:AB_2534017) secondary antibodies used at 1:500 dilution were from Thermo Fisher Scientific (Waltham, MA, USA). For isotype control, the primary antibody was omitted. Fluorescent microscopy was carried out either on an AxioObserver D1 inverted fluorescent microscope (Zeiss, Jena, Germany) or a TCS SP5 confocal microscope (Leica, Wetzlar, Germany) with 20×, 40×, and 63× lenses. Digital images were processed with Fiji academic freeware (RRID:SCR_002285) [37]. All quantifications of cellular populations were obtained by counting at least five random non-overlapping tumor-containing fields of view per section. Bright field immunohistochemistry was done as described previously [10,21,22], and the following antibodies were used: rabbit anti-versican (1:100; E-AB-36300; Elabscience, Wuhan, China), and mouse secondary anti-rabbit (1:5000; ab191866; Abcam, London, UK; RRID:AB_2650595). All quantifications of cellular populations were obtained by counting at least five random non-overlapping tumor-containing fields of view per section.

### 2.13. Bone Marrow Transfer (BMT) and Liposomal Clodronate

For adoptive BMT experiments described in detail elsewhere [10,11,20], wild-type (*WT*) and *NF-κΒ.eGFP.LUC* (*NGL*) recipient mice on the *C57BL/6* background received total body irradiation (1100 Rad) followed 12 h later by 10^7^ intravenous (via retro-orbital injection) whole bone marrow cells obtained from *WT* and *NGL* donors. One irradiated mouse per group was not transplanted with BMT to control for effective elimination of endogenous bone marrow and died 5–15 days post-irradiation. After one month, allowing for complete bone marrow reconstitution by chimeric bone marrow cells, liposomal clodronate was prepared as described previously [25,38] and 500 μg were administered intrapleurally. After yet another month required for the replacement of pleural myeloid cells by transplanted bone marrow cells [38], the mice were injected with tumor cells.

### 2.14. Bone Marrow Derived Macrophages (BMDM) 

For BMDM generation, 10^7^ bone marrow cells were plated and cultured for 7 days in the presence of 100 ng/mL macrophage colony stimulating factor (M-CSF). Where appropriate, at day 6 of the culture, recombinant human versican (1 nM) was added to the culture medium or, alternatively, the culture medium was removed and BMDM were exposed to cancer cell-conditioned media for 4 h. Culture supernatants were then isolated for ELISA and cells were processed for western blot, flow cytometry, or qPCR.

### 2.15. Immunoblotting

Nuclear and cytoplasmic protein extracts were prepared using the NEPER Extraction Kit (Thermo Fisher Scientific, Waltham, MA, USA), separated by SDS-PAGE and electroblotted to PVDF membranes (Merck Millipore, Darmstadt, Germany). Membranes were probed with the following primary antibodies: anti-IKKα (1:1000 dilution; 2682; Cell Signaling, Danvers, MA, USA; RRID:AB_331626), anti-IKKβ (1:1000 dilution; 2684; Cell Signaling; RRID:AB_2122298), anti-VCAN (1:200 dilution; ab19345; Abcam, London, UK; RRID:AB_444865), anti-β-actin (1:500 dilution; sc-47778; Santa Cruz, Dallas, TX, USA; RRID:AB_2714189), and anti-α-tubulin (TUBA; 1:4000 dilution; T5168; Sigma-Aldrich, St. Louis, MO, USA; RRID:AB_477579), followed by incubation with secondary goat anti-mouse (1:8000 dilution; 1030-05; Southern Biotech, Birmingham, AL, USA; RRID:AB_2619742) or goat anti-rabbit (1:8000 dilution; 4030-05; Southern Biotech; RRID:AB_2687483) HRP-conjugated antibodies. Membranes were visualized by chemiluminescent film exposure after incubation with enhanced chemiluminescence substrate (Merck Millipore, Darmstadt, Germany).

### 2.16. qPCR and Microarrays

Triplicate cultures of 10^6^ cells were subjected to RNA extraction using Trizol (Thermo Fisher Scientific, Waltham, MA, USA) followed by column purification and DNA removal (RNeasy Mini Kit, Qiagen, Hilden, Germany). Pooled RNA (5 μg) was quality tested on an ABI 2000 bioanalyzer (Agilent Technologies, Sta. Clara, CA, USA), labelled, and hybridized to GeneChip Mouse Gene 2.0 ST arrays (Affymetrix, Sta. Clara, CA, USA). All data were analyzed on the Affymetrix Expression and Transcriptome Analysis Consoles (RRID:SCR_018718). RNA was reverse transcribed with Superscript III (Thermo Fisher Scientific) and qPCR was performed using first-strand synthesis and SYBR FAST qPCR Kit (Kapa Biosystems, Wilmington, MA, USA) in a StepOne cycler (Applied Biosystems, Carlsbad, CA, USA). Primers for qPCR are listed in Appendix A. Ct values from triplicate reactions were analyzed with the relative quantification method 2^−ΔΔCT^ relative to mouse *Gusb* or human *ACTB* transcripts [39].

### 2.17. Shotgun Proteomics 

Supernatants obtained from murine *Kras*^MUT^ (LLC, MC38, AE17) and *Kras*^WT^ (B16F10 and PANO2) cell cultures (pooled triplicate cultures for each cell line; 5 million cells/175 cm^2^ culture flask/24 h in full DMEM followed by 24 h in FBS-free DMEM) were analyzed. For this, 600 μL of cell culture supernatant were enzymatically digested using a modified filter-aided sample preparation (FASP) protocol [40,41]. Peptides were stored at −20 °C until mass spectrometry (MS) measurements. MS data were acquired in data-dependent acquisition (DDA) mode on a Q Exactive (QE) high field (HF) mass spectrometer (Thermo Fisher Scientific). Approximately 0.5 μg per sample were automatically loaded to the online coupled RSLC (Ultimate 3000, Thermo Fisher Scientific) HPLC system. A nano trap column was used (300 μm inner diameter (ID) × 5 mm, packed with Acclaim PepMap100 C18, 5 μm, 100 Å (LC Packings, Sunnyvale, CA, USA) before separation by reversed phase chromatography (Acquity UPLC M-Class HSS T3 Column 75 µm ID × 250 mm, 1.8 µm; Waters, Eschborn, Germany) at 40 °C. Peptides were eluted from 3% to 40% over a 95 min gradient. The MS spectrum was acquired with a mass range from 300 to 1500 m/z at resolution 60,000 with AGC set to 3 × 10^6^ and a maximum of 50 ms IT. From the MS pre-scan, the 10 most abundant peptide ions were selected for fragmentation (MSMS) if at least doubly charged, with a dynamic exclusion of 30 s. MSMS spectra were recorded at 15,000 resolution with AGC set to 1 × 10^5^ and a maximum of 100 ms IT. CE was set to 28 and all spectra were recorded in profile type. Label-free quantification of DDA-MS data was performed with the Proteome discoverer (version 2.3; Thermo Fisher Scientific) using Sequest HT (as node in PD) and searching against the UniProtKB/Swiss-Prot Mouse database (release 2017_2, 16872 sequences). Searches were performed with a precursor mass tolerances of 10 ppm and fragment mass tolerances of 0.02 Da. Carbamidomethylation (C) was set as static modification, deamidation (N,Q), oxidation (M), and N-terminal Met-loss+Acetyl were selected as dynamic modifications and two missed cleavages were allowed. Percolator [42] was used for validating peptide spectrum matches and peptides, accepting only the top-scoring hit for each spectrum, and satisfying the cut-off values for FDR < 1%, and posterior error probability < 0.01. The final list of proteins complied with the strict parsimony principle. The quantification of proteins, after precursor recalibration, was based on abundance values (area under curve) for unique peptides. Abundance values were normalized in a retention time-dependent manner. The protein abundances were calculated summing the abundance values for admissible peptides. Comparisons between *Kras*^MUT^ (LLC, MC38, AE17) and *Kras*^WT^ (B16F10 and PANO2) cell lines were done using only the proteins detected in all five cell lines. 

### 2.18. Cellular Treatments 

Cells were exposed to tumor-conditioned media diluted 1:1 in DMEM. Bortezomib pre-treatment was applied 1 h prior to exposure to conditioned media at 1 μg/mL (equivalent to 3 μM). Cells were exposed to potential NF-κΒ ligands at the following concentrations: lipopolysaccharide, LPS, 1 μg/mL (equivalent to 10–20 nM); secreted phosphoprotein 1, SPP1, 100 ng/mL (equivalent to 1.25–2.5 nM); tumor necrosis factor, TNF, 20 ng/mL (equivalent to 1 nM); versican, VCAN, 360 ng/mL (equivalent to 1 nM); interleukin (IL)-1β, 30 ng/mL (equivalent to 1 nM); and C-C-motif chemokine ligand 2, CCL2, 20 ng/mL (equivalent to 1.5 nM), and were imaged for bioluminescence or processed for other assays after 4 h. The p*NGL* RAW264.7 macrophages were exposed to 1 nM VCAN followed by treatment with increasing concentrations of TLR1/TLR2 antagonist Cu-CPT22.

### 2.19. Mouse Treatments 

The IL-1 receptor antagonist isunakinra [18] was given via daily intraperitoneal injections of 20–50 mg/kg drug diluted in 100 μL PBS. Therapy was initiated at 10–17 days post s.c. tumor cells or at 5 days post-intrapleural tumor cells, allowing for efficient tumor take and a therapeutic study design. Treatment with the TLR1/TLR2 antagonist Cu-CPT22 [19] was initiated 3 days after the intrapleural cancer cell injection and consisted of daily intraperitoneal injections of 100 μL corn oil containing 10% DMSO or 20 mg/kg Cu-CPT22 diluted in 100 μL corn oil containing 10% DMSO.

### 2.20. Data Availability 

Microarray data generated during this study (GEO datasets GSE94847, GSE94880, GSE130624, and GSE130716) or published previously (GEO datasets GSE43458 and GSE103512), as well as proteomic data generated for this study (PXD019883) are available at https://www.ncbi.nlm.nih.gov/gds (accessed on 15 March 2023) and https://www.ebi.ac.uk/pride/ (accessed on 15 March 2023). Survival data were obtained from the Kaplan-Meier plotter pan-cancer RNA-seq dataset (https://kmplot.com/analysis/ (accessed on 15 March 2023)) using the search term VCAN. TCGA pan-cancer data were downloaded from https://www.cbioportal.org/ (accessed on 15 March 2023).

### 2.21. Transcription Factor Binding Site Analyses

We downloaded the RELA and RELB binding sequence motifs from the ENCODE portal [43] with the identifiers: ENCFF507YCV (CHIP-seq on HuH-7.5 cells) and ENCFF615HZF (CHIP-seq on 8988T cells), respectively, and queried the ChIPseq datasets from the ChIP-X Enrichment Analysis (CHEA) Transcription Factor Targets dataset [44,45,46]. 

### 2.22. Statistics

Sample size was calculated using power analysis on G*power [47], assuming *α* = 0.05, *β* = 0.05, and effect size *d* = 1.5. No data were excluded from analyses. Pooled data from repeated in vivo experiments are shown. All in vitro experiments were repeated independently at least three times and the stated *n* always reflects the biological and not technical sample size. Animals were allocated to treatments by randomization (when *n* ≥ 20) or alternation (when *n* < 20) and transgenic animals were enrolled case-control-wise. Data were collected by at least two blinded investigators from samples coded by non-blinded investigators. All data were tested for normality of distribution by the Kolmogorov–Smirnov test, are given as violin plots or mean ± SD, and sample size (*n*) always refers to the biological and not technical replicates. Differences in frequency were examined by Fischer’s exact and *χ*^2^ tests, in medians by Mann–Whitney or Kruskal–Wallis tests with Dunn’s post-tests, and in means by *t*-test or one-way ANOVA with Bonferroni post-tests. Changes over time and the interaction between two variables were examined by two-way ANOVA with Bonferroni post-tests. Hypergeometric tests were done at the Graeber Lab website [48]. All probability (*p*) values are two-tailed and were considered significant when *p* < 0.05. All analyses and plots were done on Prism v8.0 (GraphPad, La Jolla, CA, USA; RRID:SCR_002798).

## 3. Results

### 3.1. Non-Oncogene Addiction of KRAS-Mutant Human and Murine Cancers to IL-1β

Puzzled by the negative results of the CANOPY-2 trial, we focused on published mutation data from incident LUAD from the CANTOS trial [49] and cross-examined them with the cancer genome atlas (TCGA) LUAD dataset [50], hypothesizing that IL-1β neutralization with canakinumab would specifically prevent the development of incipient *KRAS*-mutant (^MUT^) LUAD. Indeed, *KRAS*, but not *TP53*, *EGFR*, and *BRAF*, mutations were statistically significantly under-enriched in CANTOS versus TCGA patients (Figure 1A,B). We next analyzed TCGA pan-cancer transcriptome data to discover that *IL1B* mRNA levels were elevated in *KRAS*^MUT^ and amplified cancers, and performed IL-1β immunohistochemistry in our own patients with resected LUAD [13] to find increased IL-1β protein expression in *KRAS*^MUT^ LUAD compared with *KRAS*-wild-type (^WT^) LUAD and adjacent lung tissues (Figure 1C,D). We next injected *C57BL/6* mice competent (*WT*) and diploinsufficient for *Il1b* alleles (*Il1b−/−*) [32] with syngeneic cancer cell lines carrying *Kras*^WT^ and *Kras*^MUT^ alleles [10,11]. Both subcutaneous (s.c.) and pleural routes of tumor cell injection were employed, since we previously identified that malignant pleural effusions (MPE) in mice are exclusively elicited by *Kras*^MUT^ tumor cells [10,11]. All cell lines were verified for *Kras*, *Mycoplasma spp.*, and identity status multiple times during these investigations (Appendix A). These experiments showed that specifically *Kras*^MUT^ tumors were dependent on host IL-1β (Figure 1E). Taken together, these results show that IL-1β neutralization prevents the development of incipient *KRAS*^MUT^ LUAD in humans, that *KRAS*^MUT^ human cancers contain elevated IL-1β levels, and that mouse *Kras*^MUT^ cancers are specifically dependent on host IL-1β signaling, supporting the hypothesis of a selective non-oncogene addiction of *KRAS*^MUT^ cancers to IL-1β.

### 3.2. Tumor-Associated Macrophages as a Source of Tumorigenic IL-1β

We next investigated the source of increased IL-1β in *KRAS*^MUT^ cancers, since both the host immune and tumor cells are capable of IL-1β production [20,53,54]. We were also, based on previous work, documenting that the IL-1β promoter lies under transcriptional control of NF-κB [55], a fact we validated in ChIPseq datasets from the ChIP-X Enrichment Analysis (CHEA) dataset [46] and the ENCyclopedia Of DNA Elements (ENCODE) portal [44] (Appendix A). For this, we first searched TCGA pan-cancer transcriptomes (*n* = 10,071) for associations between mRNA levels of *IL1B* and established cancer and immune cellular lineage markers. *IL1B* mRNA levels were not correlated with mRNA levels of the neutrophil marker *ELANE*, the mast cell marker *KIT*, the fibroblast marker *ACTA2*, and the endothelial marker *F8*, were significantly associated with mRNA levels of *KRAS* per se, of the pan-lymphocyte marker *CD3D*, and the cancer cell marker *KRT18*, but showed the tightest correlation (coefficient = 0.4; *p* < 10^−300^) with mRNA levels of the macrophage marker *ADGRE1* (Appendix A). To further test this, we sought to identify the host cells that respond to *KRAS*^MUT^ tumor cells with NF-κB activation, since the transcription factor controls IL-1β transcription [55] and is central to innate immune responses [56]. For this, we initiated in vivo screens of murine tumor cell lines with known *Kras* mutation status (Figure 2A and Appendix A [10]) by transplanting them into two strains of bioluminescent NF-κB reporter mice expressing ubiquitous HIV-LTR.Luciferase (*HLL* mice) [24] or NF-κB.GFP.Luciferase (*NGL* mice) [25] transgenes. Pleural injections were selected for tumor cell inoculation because they generate MPE with overt cancer-induced inflammation [10,11,20]. Serial imaging showed time-dependent NF-κB activation in host cells of recipient mice, conditional on the presence of *Kras* mutations in tumor cells (Figure 2B–E and Appendix A). The NF-κB reporter signal was emitted from pleural tumors and fluid, both containing cancer and immune cells (Figure 2F and Appendix A) [10,11,20]. Histologic and flow cytometric analysis and quantification localized the NF-κB reporter signal to tumor-infiltrating macrophages of mice with *Kras*^MUT^ pleural tumors and effusions (Figure 2G–I and Appendix A). Mast cells that foster MPE development [20] were not involved in the observed NF-κB response (Appendix A). Time-dependent NF-κB activation in host cells was stronger in pleural compared with s.c. tumor models, and required expression of mutant *Kras* by tumor cells (Appendix A). Adoptive bone marrow transfer corroborated myeloid cells as the origin of tumor-induced NF-κB activation, and pharmacologic killing of pleural macrophages prevented host NF-κB activation and pleural carcinomatosis (Appendix A). 

The pro-tumor function of pleural macrophages was also consistent with the phenotype of macrophage-depleted *Lyz2.Cre;Dta* mice [21] (Appendix A). Tumor-secreted solute factors are responsible for NF-κB activation in macrophages, since murine RAW264.7 macrophages stably expressing the *NGL* reporter responded with robust in vitro NF-κB activation to cell-free media conditioned by *Kras*^MUT^, but not by *Kras*^WT^ or *Kras*-silenced, tumor cells (Figure 3A,B and Appendix A). This NF-κB response requires canonical NF-κB signaling, since it involved IKKβ and was attenuated by the proteasome inhibitor bortezomib (Figure 3C,D, Appendix A). Proteasome-dependent canonical NF-κB activity was also documented in bone marrow-derived macrophages (BMDM) derived from *NGL* mice (Figure 3E,F). Differential gene expression (ΔGE) analyses (GEO datasets GSE94847, GSE94880, GSE130624, and GSE130716; total *n* = 32) identified 13 BMDM-specific transcripts that were further induced by incubation with tumor-conditioned media (ΔGE > 5; ANOVA *p* < 0.05) and included *Il1b* but not *Il6* and *Tnf* reported elsewhere [57] (Figure 3G and Appendix A). In addition, *NGL* mice diploinsufficient in *Il1b* alleles [32] were resistant to tumor-induced NF-κB activation (Figure 3H). Incubation of BMDM with *Kras*^MUT^ tumor-conditioned media promoted their differentiation as assessed by flow cytometry for markers MHCII and CD206, and *Il1b* mRNA and IL-1β protein expression (Figure 3I–L). These data directly show that *KRAS*^MUT^ tumor cells can activate NF-κB in macrophages via solute mediator(s) that trigger IKKβ-mediated NF-κB activation, differentiation, and IL-1β elaboration.

### 3.3. Tumor-Secreted Versican as a Key Macrophage Effector

We next compared *Kras*^MUT^ with *Kras*^WT^ cancer cells for secretory molecules triggering macrophage NF-κB activation. Microarrays identified 25 transcripts over-represented in *Kras*^MUT^ tumor cells, and a proteomic screen of tumor cell-conditioned media detected 226 proteins secreted > 10-fold by *Kras*^MUT^ over *Kras*^WT^ cells, with the glycoprotein versican (VCAN; encoded by the human/murine *VCAN/Vcan* genes) emerging from both screens and withstanding validation (Figure 4A–E and Appendix A). Multiple NF-κΒ ligands were also screened using *pNGL*-expressing RAW264.7 macrophages, revealing that the toll-like receptor (TLR)2 ligand VCAN potently activates macrophage NF-κΒ-driven transcription to the same degree as the TLR4 ligand lipopolysaccharide (LPS) (Figure 4F,G). VCAN also induced IKKβ in primary murine BMDM, which were verified by microarray to overexpress > 10-fold over cancer cells TLR1, TLR2, TLR6-9, and TLR13 (Figure 4H, Appendix A). Importantly, shRNA-mediated *Vcan* silencing in LLC cells diminished their ability to trigger NF-κB activation in *NGL* mice and to precipitate MPE (Figure 4I–M and Appendix A). VCAN overexpression is not restricted to mouse *Kras*^MUT^ cancers, since *VCAN* transcripts are also over-represented in human cancers with high *KRAS*^MUT^ frequencies (derived from the catalogue of somatic mutations in cancer, COSMIC), such as LUAD from smokers (GEO dataset GSE43458), and NSCLC and colorectal adenocarcinoma (COAD/READ; GEO dataset GSE103512) (Appendix A) [58,59,60]. High *VCAN* mRNA expression also portended poor survival in a number of human cancers from the KMplot pan-cancer RNAseq dataset (Appendix A) [61]. Analysis of samples from two of our own clinical studies [13,14] showed that VCAN protein expression was significantly increased in LUAD compared with adjacent lung tissues and that *VCAN* mRNA expression was significantly increased in human MPE compared with benign pleural effusions (BPE) (Figure 4N,O). To test whether the proposed inflammatory loop can serve as a diagnostic tool to distinguish MPE from BPE, which is an unmet clinical need [14], *pNGL*-expressing RAW264.7 macrophages were exposed to cell-free supernatants from human pleural effusions. After 4 h, a robust NF-κB reporter signal was triggered selectively by MPE supernatants (Figure 4P). Taken together, these data indicate that VCAN secreted by cancer cells triggers IKKβ-mediated NF-κB activation in tumor-associated macrophages and promotes metastasis. Moreover, VCAN is overexpressed in human *KRAS*^MUT^ cancers and can serve as a diagnostic and prognosis biomarker.

### 3.4. Myeloid IKKB as the VCAN Accessory

To identify the IKK responsible for NF-κB signaling in macrophages, we silenced the four main IKKs (encoded by the murine *Chuk*, *Ikbkb*, *Ikbke*, and *Tbk1* genes) in RAW264.7 macrophages and identify IKKβ as the main mediator of NF-κB activation in these cells (Figure 5A,B). To further define myeloid IKKβ functions, we obtained BMDM from intercrosses of *Lyz2.Cre* mice with mice carrying conditionally-deleted alleles of IKKα (*Chuk^f/f^*) and IKKβ (*Ikbkb^f/f^*), as well as with *Cre*-reporter mice switching from red to green fluorescence upon *Cre*-mediated recombination (*mT/mG*), all reported previously [21,22]. Treatment of bone marrow cells from *mT/mG;Lyz2.Cre* mice with macrophage-colony stimulating factor (M-CSF; 100 ng/mL) to drive them towards macrophage differentiation and lysozyme 2 (LYZ2) expression yielded efficient *Cre*-mediated recombination (Figure 5C). Flow cytometric assessment of BMDM derived from these mice showed that intact IKKβ signaling in primary macrophages is essential for their differentiation and expression of critical pro-inflammatory genes including *Lyz2*, *Il1b*, and *C3* (Figure 5D–F and Appendix A). Finally, two different syngeneic *Kras*^MUT^ tumor cell lines featuring VCAN overexpression were inoculated into the pleural space of the above myeloid IKK-deleted mice, to reveal that intact IKKβ signaling in macrophages is required for MPE (Figure 5G). Thus, VCAN-driven IKKβ activation mediates NF-κB signaling, IL-1β expression, differentiation, and pro-tumor function of macrophages (Figure 5H). To further query the proposed KRAS-VCAN-IKKβ connection, we interrogated mutations, copy number alterations, and fusions of the encoding genes in the TCGA pan-cancer dataset. Interestingly, *VCAN* and *IKBKB* alterations (mostly missense mutations) each occur in 5% of all cancer patients and are significantly mutually enriched (*VCAN* in *KRAS* and *IKBKB* in *VCAN* mutations) suggesting mutual addiction (Appendix A–C). In addition, *KRAS*, *IKBKB*, and *VCAN* alteration frequencies across 32 human cancer types are tightly correlated, and were highest in LUAD, COAD/READ, and uterine corpus endometrial carcinoma (UCEC), cancers that commonly cause MPE (Appendix A). In the latter tumor types featuring *KRAS*, *IKBKB*, and *VCAN* alteration frequencies, addiction of *IKBKB* and *VCAN* mutations persisted, and patients with *VCAN* and/or *IKBKB*-altered cancers displayed decreased body mass (cachexia), higher mutation burden, microsatellite instability, and hypoxia indices (Appendix A). Collectively, these data support that tumor cell VCAN cooperates with myeloid IKKβ in mouse and human cancers.

### 3.5. Non-oncogene Addiction of KRAS-Mutant Tumors to IL-1β Is Actionable

To block the proposed inflammatory loop, we employed the novel IL-1 receptor antagonist isunakinra [18]. Systemic delivery of isunakinra to mice with already established tumors specifically inhibited s.c. growth of *Kras*^MUT^ tumors (Figure 6A). In addition, isunakinra limited NF-κΒ activation in *Kras*^MUT^ cancer cells in vivo, a phenomenon we previously showed to be fueled by myeloid IL-1β, as well as their ability for lethal MPE induction (Figure 6B–D). Since VCAN is a known TLR2 ligand [57], the pro-inflammatory loop proposed here was also targeted with the TLR1/2 inhibitor Cu-CPT22 [19]. The drug effectively inhibited VCAN-induced NF-κΒ activation and cellular survival in RAW264.7 macrophages at the low micromolar range and blocked tumor growth in vivo at clinically relevant concentrations (Figure 6E–H). Hence, VCAN-IKKβ-mediated addiction of *KRAS*^MUT^ cancers to host IL-1β can be used to indirectly target these tumors. 

## 4. Discussion

Here, we show how *KRAS*-mutant tumors are dependent on IL-β provided by tumor-associated macrophages. Importantly, we show that tumor-secreted versican causes IKKβ activation in myeloid cells to foster this pro-inflammatory circuitry. Notwithstanding cancers with other mutations and other myeloid cells like neutrophils and mast cells that might also fuel tumors with IL-1β, we define here a non-oncogene addiction of *KRAS* and IL-1β, in tandem with their partners in crime VCAN and IKKβ. The findings stress the need for molecular stratification of current clinical trials of IL-1β inhibition against lung cancer. Unique experimental models for the study of tumor genome-host immunity interactions are provided, and novel diagnostic platforms and prognostic biomarkers are described for further validation.

Although sotorasib was recently approved in the U.S. against *KRAS*^G12C^-mutant NSCLC [62], *KRAS*-mutant cancers from multiple sites of origin remain notoriously aggressive and undruggable [63] and direct KRAS inhibition is associated with some toxicity that likely renders such treatments unsuitable for chemoprevention [64]. On the contrary, anti-IL-1β-directed therapies hold promise for chemoprevention, as shown by the CANTOS trial, (where tri-monthly administration of the IL-1β-neutralizing antibody canakinumab over 3.7 years of observation decreased overall and lung cancer mortality by 51% and 77%, respectively) based on their excellent safety profile [2]. The pro-inflammatory interplay between VCAN in tumor cells and IKKβ in macrophages described here is not only mechanistically intriguing, but also promising for innovations in cancer therapy and diagnosis. We identify cancer cell VCAN and myeloid IKKβ as the accomplices of KRAS that trigger secretion of IL-1β in the milieu of *KRAS*-mutant cancers. The results position these cancers as favorable candidates for anti-IL-1β therapy, and versican as a diagnostic and prognostic biomarker, as well as a therapeutic target in this tumor category that comprises 9% of all human cancers, alone or in combination with anti-IL-1β agents. In addition, since early diagnosis of metastasis is key to effective cancer therapy [57], VCAN can serve as a biomarker of metastasis. This might be achieved by monitoring local or systemic VCAN levels in patients at risk, or by using our NF-κB-reporter macrophages as a diagnostic platform. Indeed, our data indicate that the latter can accurately discriminate pleural metastasis from other pleural inflammatory processes, highlighting the clinical relevance of our findings.

NF-κB signaling in cancer and myeloid cells impacts modes of tumor progression and metastasis in various tumor types and is intimately addicted with oncogenic KRAS signaling [65,66]. However, the lessons learnt from clinical trials of proteasome (and hence also canonical NF-κB pathway) inhibitors against multiple myeloma dictate that therapeutic interventions into the NF-κB pathway are also associated with significant toxicity, since the pathway acts simultaneously in epithelial and immune cells in opposing fashions [67,68]. In addition to previous work elucidating the oncogenic functions of IKKβ in tumor cells [6,11,22,34,53,65,66], here we show how myeloid IKKβ functions to fuel tumor cell NF-κB signaling with IL-1β, further emphasizing the complex and multifaceted pro-tumor functions of NF-κB and the need for its therapeutic targeting against cancer.

## 5. Conclusions

In conclusion, KRAS-mutant cancers rely on host IL-1β, which they elicit from host macrophages via secretory versican that activates myeloid IKKβ. This inflammatory loop provides multiple opportunities for improved diagnosis, prognostication, and identification of therapeutic vulnerabilities of *KRAS*-mutant cancers.

## Figures and Tables

**Figure 1 cancers-15-01866-f001:**
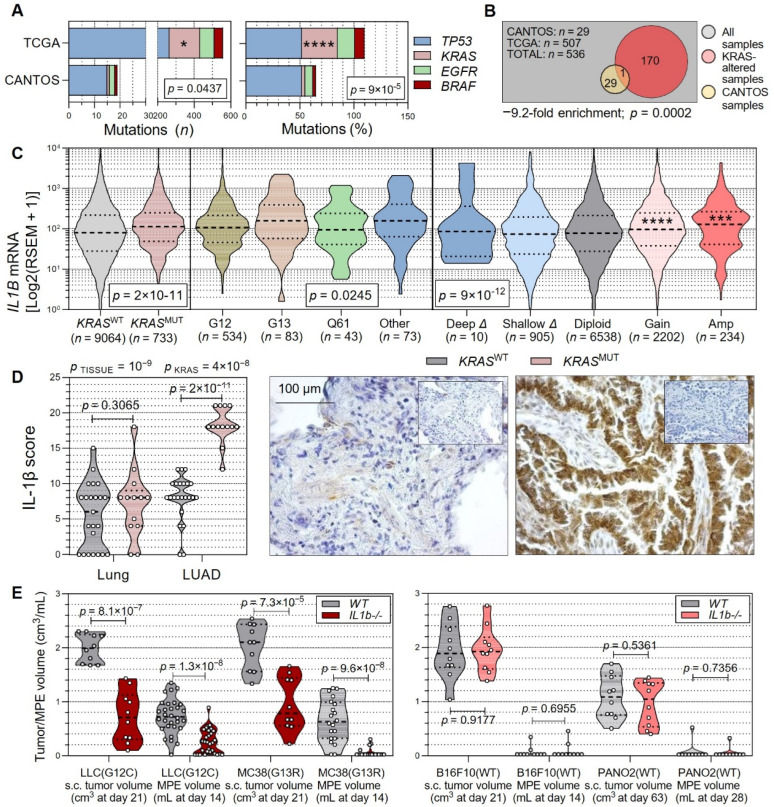
Non-oncogene addiction of *KRAS*-mutant cancers to interleukin (IL)-1β. (**A**,**B**) *TP53*, *KRAS*, *EGFR*, and *BRAF* mutation frequencies in the canakinumab anti-inflammatory thrombosis outcomes study (CANTOS) and the cancer genome atlas (TCGA) lung adenocarcinoma (LUAD) patients. Data from [2,50,51,52]. Shown are patient and mutation numbers (*n*) and percentages (%), as well as probabilities (*P*), *χ*^2^ test (**A**) or hypergeometric test (**B**). (**C**) Data summary of *KRAS* alterations (G12, G13, Q61) versus *IL1B* mRNA expression in the cancer genome atlas (TCGA) pan-cancer dataset (*n* = 10,967 samples from 10,953 patients with 31 different cancers) from the US. Data from [51,52]. RSEM, RNA-Seq by Expectation-Maximization. Note the elevated *IL1B* mRNA expression of *KRAS*-altered cancers. (**D**) Data summary (left) and representative images (right; inlays: isotype controls) of *KRAS* alterations versus IL-1β protein expression in lung adenocarcinoma (LUAD) and adjacent lung tissue from *n* = 36 resected patients from Munich, Germany. Note the elevated IL-1β protein expression of *KRAS*-altered LUAD. (**E**) Data summary of subcutaneous (s.c.) tumor and malignant pleural effusion (MPE) volume of *C57BL/6* mice competent (*WT*) or diploinsufficient (*Il1b-/-*) in *Il1b* alleles at the indicated time-points after s.c. or intrapleural injection of 5 or 2 × 10^5^ tumor cells, respectively, with (left; *n* from left to right = 10, 10, 30, 30, 10, 10, 20, and 20) or without (right; *n* = 10/group) *Kras* mutations. Note the requirement of *Kras*-altered tumors for host IL-1β. Shown are raw data (circles), rotated kernel density distributions (violins), medians (dashed lines), quartiles (dotted lines), and *p*, probabilities, Kolmogorov–Smirnov or Kruskal–Wallis test (**A**), two-way ANOVA (**B**, above graph) and Bonferroni post-tests (**B**, in graph), or unpaired *t*-tests (**C**). *, *** and ****: *p* < 0.05, *p* < 0.001 and *p* < 0.0001, respectively, compared with diploid patients, Dunn’s post-tests.

**Figure 2 cancers-15-01866-f002:**
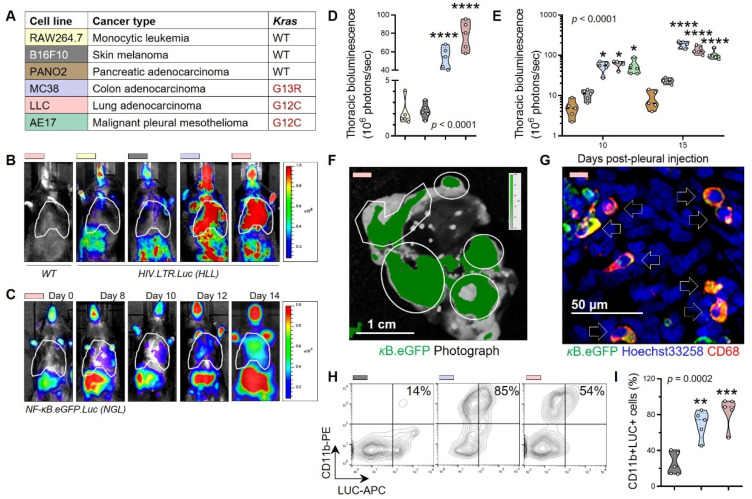
*Kras*-mutant tumors activate NF-κB in tumor-infiltrating macrophages. (**A**) Color-coded cancer cell lines used with tissues of origin and *Kras* mutation status. (**B**–**E**) Bioluminescent images with pseudo color scales (**B**,**C**) and data summaries (**D**,**E**) from *WT* and HIV-LTR.Luciferase (*HLL*: **B** and **D**), and NF-κB.GFP.Luciferase (*NGL*; **C** and **E**) NF-κB reporter mice at 14 days (**B**,**D**) or serial time-points (**C**,**E**) post-pleural injection of tumor cells. Note that in these models, bioluminescence is exclusively emitted by the host and not the tumor cells. (**B**,**C**) Dashed areas delineate the thorax. (**F**) Photographic/biofluorescent image overlay with pseudo color scale of *NGL* mouse lung explant 14 days post-pleural LLC cells shows NF-κB reporter GFP signal (κB.eGFP) over pleural tumors (outlines; *n* = 10). (**G**) GFP immunoreactivity of pleural tumor sections co-localizes with the macrophage marker CD68 (arrows; *n* = 10). (**H**,**I**) Flow cytometric contour plots (**H**) and data summary (**I**) of pleural tumor cells from *NGL* mice obtained 14 days post-pleural injection stained for the myeloid marker CD11b and the κB.LUC reporter. Percentages in (**H**) pertain to CD11b+LUC+ cells. Data in (**D**,**E**,**I**) are given as raw data (circles), median (dashed lines), quartiles (dotted lines), and kernel density distributions (violin plots) color-coded as in (**A**). Sample size (*n*) = 5–10/group; *p*, probability, one- or two-way ANOVA; *, **, ***, and ****, *p* < 0.05, *p* < 0.01, *p* < 0.001, and *p* < 0.0001, respectively, compared with mice injected with RAW264.7, PANO2, or B16F10 cells at the same time-points, Bonferroni post-tests.

**Figure 3 cancers-15-01866-f003:**
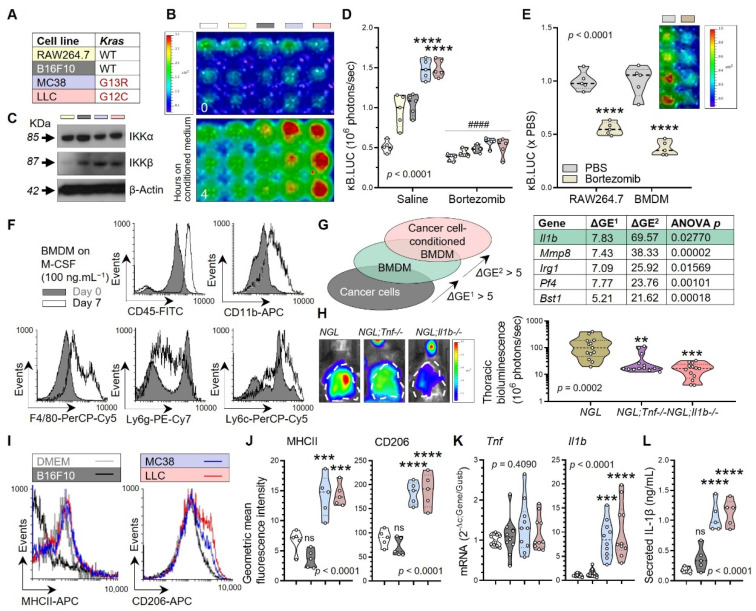
Tumor-secreted factors drive IKKβ activation, differentiation, and IL-1β secretion in macrophages. (**A**) Color-coded cancer cells with *Kras* mutation status. (**B**–**E**) Bioluminescent images with pseudo color scale (**B**,**E**), immunoblots (**C**), and data summaries (**D**,**E**) from exposure of RAW264.7 macrophages stably expressing p*NGL* (**B**–**D**) and murine bone marrow-derived macrophages (BMDM) obtained from *NGL* mice after one-week 100 ng/mL M-CSF exposure (**E**) to cell-free tumor-conditioned media or DMEM (white boxes) with or without bortezomib pretreatment (1 μg/mL~3 μM for 1 h). (**D**,**E**) *n* = 5/group; *P*, probability, one- or two-way ANOVA; **** and ^####^, *p* < 0.0001 compared with other groups or saline-treated cells, respectively, Bonferroni post-tests. (**F**) Flow cytometry-assessed differentiation marker expression of murine bone marrow cells before (day 0) and after (day 7) one-week M-CSF exposure. (**G**) Microarray strategy and top-five differentially expressed genes (ΔGE) of murine BMDM compared with cancer cells (ΔGE^1^) and of tumor-conditioned BMDM compared with naïve BMDM (ΔGE^2^). *n* = 5/group; *P*, probability, one-way ANOVA. (**H**) Bioluminescent images and data summary of *NGL*, *NGL; Tnf−/−*, and *NGL; Il1b−/−* mice 14 days post-pleural injection of LLC cells. *n* = 13/group; *P*, probability, one-way ANOVA; ** and ***, *p* < 0.01 and *p* < 0.001, respectively, compared with *NGL* mice, Bonferroni post-tests. (**I**–**L**) Histograms (**I**) and data summaries (**J**–**L**) of naïve or tumor-conditioned BMDM for macrophage differentiation markers (**I**,**J**), *Tnf* and *Il1b* mRNA (**K**), and IL-1β protein (**L**) expression. *n* = 5–10/group; *P*, probability, one-way ANOVA; *** and ****, *p* < 0.001 and *p* < 0.0001, respectively, compared with DMEM and B16F10-conditioned media, Bonferroni post-tests. Data are given as raw data (circles), medians (dashed lines), quartiles (dotted lines), and kernel density distributions (violin plots) color-coded as in (**A**). The uncropped blots are shown in Appendix A.

**Figure 4 cancers-15-01866-f004:**
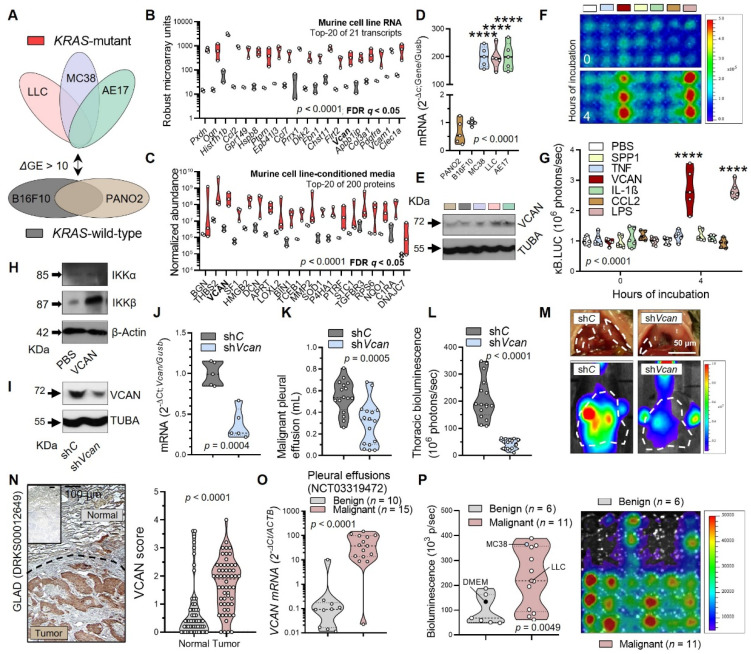
Tumor-secreted versican drives macrophage IKKβ, metastasis, and is a cancer biomarker. (**A**–**E**) *Kras*-mutant and wild-type cancer cell RNA and supernatants were subjected to microarray and LC-MSMS analyses, respectively. Shown are the experimental design (**A**), top-20 over-represented transcripts (**B**) and secretory proteins (**C**), and *Vcan*/VCAN mRNA/protein expression (**D**,**E**). *n* = 2–3/group; *P*, probability, two-way ANOVA; Bold letters, false discovery rate (FDR) *q* < 0.05 compared with *Kras*-wild-type cells, two-stage linear step-up procedure of Benjamini, Hochberg, and Yekutieli. (**F**,**G**) Representative bioluminescent images (**F**) and data summary (**G**) from p*NGL* RAW264.7 cells exposed to lipopolysaccharide (LPS; 1 μg/mL) or recombinant proteins (1–2 nM). *n* = 5/group; *P*, probability, two-way ANOVA; ****, *p* < 0.0001 compared with other groups, Bonferroni post-tests. (**H**) Immunoblots of mouse BMDM exposed to VCAN. *n* = 5/group. (**I**–**M**) LLC cells stably expressing control (sh*C*) and anti-*Vcan* (sh*Vcan*) shRNA were validated and injected intrapleurally into *NGL* mice. Shown are immunoblots (**I**), *Vcan* mRNA expression (**J**), and data summaries (**K**,**L**) and representative photographic and bioluminescent images (**M**) taken 14 days post-tumor cells. (**I**,**J**) *n* = 5/group; (**K**–**M**) *n* = 16/group; *P*, probability, unpaired Student’s *t*-test. (**N**) Images and data summary of VCAN expression of *n* = 41 tumor/normal tissue pairs from patients with resected lung adenocarcinoma. (**O**) *VCAN* mRNA expression of 10 benign and 15 malignant pleural effusions. (**P**) Data summary and representative image of bioluminescence of p*NGL* RAW264.7 cells after exposure to benign (*n* = 6; top triplicates) and malignant (*n* = 11; bottom triplicates) pleural effusions and tumor-conditioned media (each triplicate column is one patient). (**N**–**P**) *P*, probability, unpaired Student’s *t*-test. (**B**–**D**,**G**,**J**–**L**,**N**–**P**) Shown are raw data (circles), kernel density distributions (violins), and medians/quartiles (dashed/dotted lines). The uncropped blots are shown in Appendix A.

**Figure 5 cancers-15-01866-f005:**
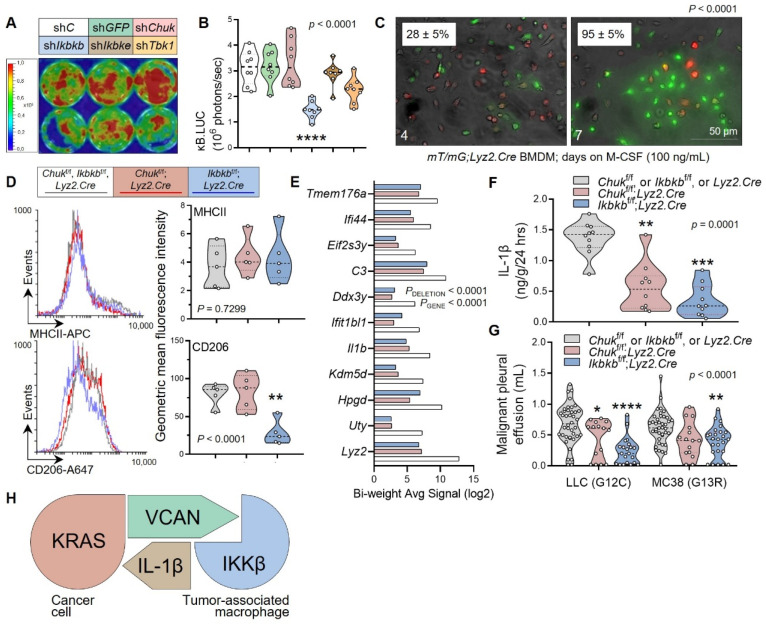
IKKβ mediates pro-tumor NF-κΒ activity, differentiation, and IL-1β secretion in macrophages. (**A**,**B**) Bioluminescent image with pseudo color scale (**A**) and data summary (**B**) of p*NGL* RAW264.7 cells 72 h post-infection with control (sh*C*), anti-GFP (sh*GFP*), or anti-inhibitor of NF-κB kinase (sh*Chuk*, sh*Ikbkb*, sh*Ikbke*, or sh*Tbk1*)-specific shRNAs. *n* = 8 independent experiments/group; *P*, probability, one-way ANOVA; ****, *p* < 0.0001 compared with sh*C*, Bonferroni post-tests. (**C**–**F**) Bone marrow-derived macrophages (BMDM) were derived from *mT/mG; Lyz2.Cre*, *Chuk*^f/f^*; Lyz2.Cre*, and *Ikbkb*^f/f^*; Lyz2.Cre* mice using one-week exposure to 100 ng/mL M-CSF. Shown are images and mean ± SD % green cells of bone marrow cells from *mT/mG; Lyz2.Cre* mice during/after weekly treatment with M-CSF (**C**), flow cytometric histograms (left) and data summary (right) of marker expression (**D**), top-differentially expressed genes by microarray (**E**), and interleukin (IL)-1β secretion by ELISA (**F**). (**C**,**D**) *n* = 5 independent experiments/group; *P*, probability, Fisher’s exact test or one-way ANOVA; **, *p* < 0.01 compared with other groups, Bonferroni post-tests. (**E**) *n* = 1 pooled triplicate/group; *P*, probabilities, two-way ANOVA. (**F**) *n* = 10 independent experiments; *P*, probability, one-way ANOVA; ** and ***, *p* < 0.01 and *p* < 0.001, respectively, compared with controls, Bonferroni post-tests. (**G**) *Chuk*^f/f^*; Lyz2.Cre* and *Ikbkb*^f/f^*; Lyz2.Cre* mice received intrapleural LLC or MC38 cells, and were evaluated after 14 days for malignant pleural effusions (MPE). Data summary of *n* = 40, 15, and 21 single transgenic control, *Chuk*^f/f^; *Lyz2.Cre*, and *Ikbkb*^f/f^; *Lyz2.Cre* mice injected with LLC cells, respectively, and of *n* = 40, 15, and 25 respective mice injected with MC38 cells. *P*, probability, two-way ANOVA; *, **, and ****, *p* < 0.05, *p* < 0.01, and *p* < 0.0001, respectively, compared with controls, Bonferroni post-tests. (**H**) Schematic of the proposed mechanism for non-oncogene addiction of *KRAS*-mutant cancers to IL-1β. To this end, *KRAS*-mutant cancers secrete VCAN to co-opt IKKβ in macrophages within the metastatic niche, which drives IL-1β secretion by macrophages to foster tumor progression.

**Figure 6 cancers-15-01866-f006:**
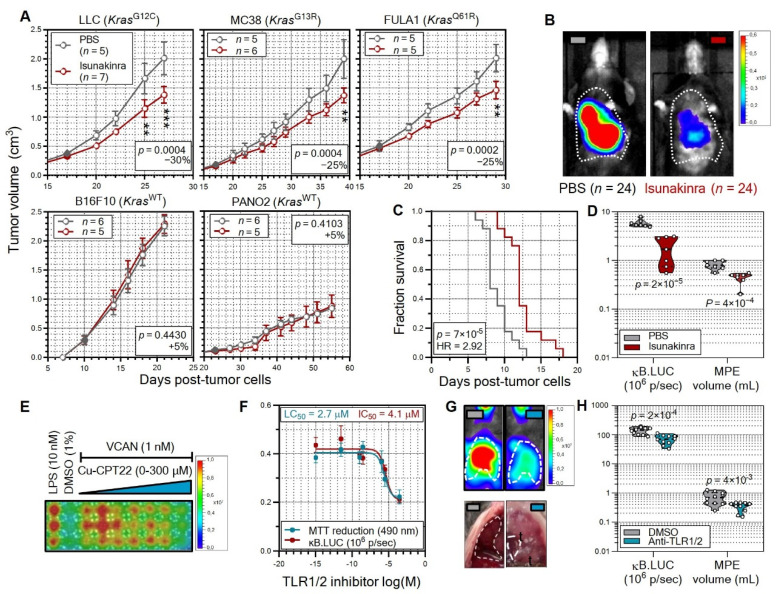
Pharmacologic abolition of non-oncogene addiction of *Kras*-mutant tumors to IL-1β. (**A**–**D**) The IL-1 receptor antagonist isunakinra limits nuclear factor (NF)-κΒ activation and tumor growth of *Kras*-mutant cancer cells. (**A**) *FVB* (*n* = 10) and *C57BL/6* (*n* = 45) mice received subcutaneous injections of 5 × 10^6^ FULA1 (*FVB* mice) or LLC, MC38, B16F10, or PANO2 (*C57BL/6* mice) cells that carry G12C, G13R, Q61R, or wild-type (^WT^) *Kras* alleles. Mice were allowed 10–23 days for tumor take (solid circles) and were treated with daily intraperitoneal PBS or 20 mg/Kg isunakinra until control tumor volume reached 1 cm^3^ (PANO2 cells) or 2 cm^3^ (all other cell lines). Shown are mouse numbers (*n*), tumor volume as mean (circles) and SD (bars), two-way ANOVA probability (*P*) for treatment effects, and average isunakinra effect at the last time-points (%). ** and ***: *p* < 0.01 and *p* < 0.001, respectively, Bonferroni post-tests. (**B**–**D**) *C57BL/6* mice received intraperitoneal PBS or 20 mg/Kg isunakinra (*n* = 7/group) followed 1 h later by 10^6^ intrapleural LLC cells stably expressing a κΒ.LUC reporter (*NGL*), and were imaged for bioluminescence 4 h later. Shown in (**B**) are representative chest (dotted lines) bioluminescent images with pseudo color scale. In addition, *FVB* mice (*n* = 48) received 2 × 10^5^ intrapleural FULA1 cells, were allowed 5 days for tumor take and received daily intraperitoneal PBS or 20 mg/Kg isunakinra. Mice were sacrificed when morbid for survival analyses (*n* = 17/treatment) or at day 14 post-tumor cells for malignant pleural effusion (MPE) analyses (*n* = 7/treatment). Shown in (**C**) are Kaplan–Meier survival estimates (curves) with log-rank probability (*P*) and hazard ratio (HR), and in (**D**) data summary of chest bioluminescence and MPE volume, shown as raw data points (circles), medians (dashed lines), quartiles (dotted lines), kernel density distributions (violins), and probability (*P*), unpaired Student’s *t*-test. (**E**–**H**) The toll-like receptor 1/2 (TLR1/2) inhibitor Cu-CPT22 blocks versican (VCAN)-induced myeloid NF-κΒ activation and MPE of *Kras*-mutant cancer cells in vivo. (**E**,**F**) Representative bioluminescent image with pseudo color scale (**E**) and results summary (**F**) of RAW264.7 macrophages stably expressing *NGL* that were pre-treated with 1% DMSO or increasing Cu-CPT22 concentrations in 1% DMSO and were exposed (1 h latency) to 10 nM lipopolysaccharide (LPS) or 1 nM recombinant VCAN. Cells were assessed for bioluminescence at 24 h and for MTT reduction at 72 h post-LPS/VCAN treatments. The *n* = 3 and *n* = 6 independent experiments/group for κΒ.LUC and MTT, respectively, are shown as 50% inhibitory/lethal concentrations (IC_50_/LC_50_), mean (circles), and SD (bars). (**G**,**H**) Representative images (**G**) and data summary (**H**) of chest bioluminescence and MPE volume of κΒ.luc mice at 14 days post-pleural injection of 2 × 10^5^ LLC cells followed by treatment with daily intraperitoneal injections of 100 μL corn oil containing 10% DMSO (*n* = 10) or 20 mg/kg Cu-CPT22 diluted in 100 μL corn oil containing 10% DMSO (*n* = 10) initiated 5 days post-LLC cells. t, intrapleural tumors. Shown are raw data points (circles), medians (dashed lines), quartiles (dotted lines), kernel density distributions (violins), and probability (*P*), unpaired Student’s *t*-test.

## Data Availability

Microarray data generated during this study (GEO datasets GSE94847, GSE94880, GSE130624, and GSE130716) or published previously (GEO datasets GSE43458 and GSE103512), as well as proteomic data generated for this study (PXD019883) are available at https://www.ncbi.nlm.nih.gov/gds (accessed on 15 March 2023) and https://www.ebi.ac.uk/pride/ (accessed on 15 March 2023). Survival data were obtained from the Kaplan-Meier plotter pan-cancer RNA-seq dataset [61] using search term VCAN. TCGA pan-cancer data were downloaded from [51]. All data and materials are available upon request.

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
