# Peer review of "Non-Oncogene Addiction of KRAS-Mutant Cancers to IL-1β via Versican and Mononuclear IKKβ"

_cancers, 2023, doi:10.3390/cancers15061866_

Round 1

Reviewer 1 Report

I must share my compliments with the authors for this manuscript. Starting from an unattended negative result in the CANPOY-2 trial they’ve challenged the molecular understanding of tumor environment and the role of NF-kB and IL-1b in lung adenocarcinoma.

The paper is well-written in proper English. The Material and Methods section is correct and conscious in regards of the study outline/project and ethic awareness.

In the Results Figures 1, 2, 6 are not correctly formatted.

Reviewer 2 Report

The authors show that KRAS mutant cancers secrete the protein VCAN, which can drive the activation of NF-kB in host macrophages. In turn, these macrophages can fuel the cancers with IL-1b to promote tumor growth. Overall, the data are clearly presented and will be of interest to the readers. However, I suggest several experiments that may add strength to understanding the role of VCAN in tumor-associated inflammation.

1. In Figure 4, although it was shown that VCAN knockdown can decrease MPEs and thoracic bioluminescence, data to show whether VCAN knockdown can decrease IKKb signaling or NF-kB activation in host cells of recipient mice using the NF-kB reporter HLL mice or NGL mice is important to include. This may provide direct evidence for the role of VCAN to induce NF-kB activation in host cells.

2. Similarly, although it was shown that blocking IL-1b signaling or TLR1/2 signaling in KRAS mutant cells can block in vivo growth, I believe it would be important to include data to show whether VCAN knockdown affects in vivo growth as well. As the authors are suggesting that VCAN plays a significant role in driving NF-kB activation and IL-1b expression to promote tumor growth, this is an essential piece of data.

3. Overall, the manuscript has overlapping parts with the paper "Myeloid-derived interleukin-1β drives oncogenic KRAS-NF-κΒ addiction in malignant pleural effusion". I suggest that the authors emphasize the novel findings in this manuscript which are focused on VCAN and macrophages. From my understanding, much of the data in Figures 1 and 2 can be omitted as they are overlapping with the aforementioned paper.

4. As a minor comment, Figures 1, 2, and 6 are cut at the right side and, therefore, missing parts of the data.

5. Also, inserting a line change between line 499 and 500 may help with the flow of content.

Reviewer 3 Report

The article of Magda Spella et al entitled “Non-oncogene addiction of KRAS-mutant cancers to IL-1β via versican and mononuclear IKKβ” presents research on the KRAS gene. The KRAS gene is one of the most frequently mutated proto-oncogenes in human cancers, including colorectal, pancreatic and lung cancers. Knowledge about the pathways connecting KRAS-cancer cells with IL-1β may be crucial for the treatment of this type of cancer and may determine the success of the therapy. This paper presents a carefully planned and performed research. Conclusions are presented very logically and exhaustively documented. I only have a few minor comments, but they do not detract from the high quality of this work.

1.     In many places (for example in lines 61-63, 100, 109-110, 113-117, 252, 394, 397, 413, 445, 449, 468) there are links to the websites which should be placed rather in References, because they interfere with reading the text.

2.     In the lines 120-121 there is placed chemical name of D-Luciferin potassium salt, but there are not placed chemical names for another compounds like bortezomib (line 131), puromycin (line125) or geneticin (line 127). In my opinion chemical names of known compounds are not necessary, especially when is placed CAS number, but if the authors want to place them than it should be for all of these compounds.

3.     There are some typo and editorial errors:

·        In the line 466 there is “trnscriptional" instead “transcriptional”

·        In the lines 482 and 483 there is lack of space before word “Luciferase”

·        Figures 1, 2 and 6 they are not complete, they are cut off

·        In References are double numbers

Reviewer 4 Report

The authors did a thorough study on the investigation of the pathways behind KRAS-mutant tumor growth. This article discussed the relationship between KRAS-mutant tumor growth and macrophage-secreted IL-1β and the activation of IKKβ pathway in myeloid cells by tumor-secreted versican. The authors provided adequate details in experiments design and results illustration. Overall, the manuscript did a comprehensive investigation and will attract wide interest in this field. I suggest acceptance in current version. Here are some comments to the authors. The only minor thing is that the font size/space need to be consistent across the manuscript.

Reviewer 5 Report

In the present study Spella et al elucidated an interesting KRAS mutated tumor and tumor microenvironment signalling pathways in which KRAS mutated tumor cells release the inflammatory versican signlaing to act upon TLR receptor on macrophage, the latter through IKKβ activated nuclear factor (NF)-κB resulting in overexpression of  interleukin (IL)-1β (addiction). The study is appropriately conducted and the results are nicely presented and statistical analyses are well chosen.

The authors may considered shortening the legends to figures and extending the discussion. Also the manuscript should be checked carefully for typos

Round 2

Reviewer 2 Report

.